# Wind Drift, Breakdown, and Pile Up of the Ice Field

Vadim K. Goncharov

Department Ocean Technics and Marine Technologies, Saint Petersburg State Marine Technical University, Lotsmanskaya Str. 3, 190121 Saint Petersburg, Russia; vkgonch@mail.ru

**Abstract:** This article contains the analytical model of the drift of a separate ice field under the action of wind and current, in which velocities and directions can vary over time. The model takes into account the mass of ice, added mass of seawater, and the effects of the wind and current on the ice field in forming the friction on its upper and underwater surfaces and the frontal resistance on its end (forward and backward) surfaces. Simulation of the wind drift of the ice field showed the drift velocity exceeds the considerable known velocity of a compact ice cover drift. A drifting ice field has a certain kinetic energy that should be released when a collision occurs with an unmovable obstacle, and spent on brittle breakdown of a quantity of the ice field. The volume of formed small ice pieces (fragments of ice field) was estimated by comparison of the specific energy of the sea ice brittle destruction and the kinetic energy of the drifting ice field. The article presents the results of the estimation of the possible volume of the ice pieces and the scales of formed piles as a result of a collision with an obstacle, depending on the initial dimensions of the ice field and wind speed. Developed models and the results of computer modeling can be used to estimate the ice pile sizes near the stationary platforms and terminals on the Arctic seas.

**Keywords:** breakdown; collision; current; drift; ice field; ice fragment; pile; resistance; wind

## 1. Introduction

Ice piling near the coastal and shelf structures is a characteristic phenomenon in the water areas of the freezing seas. Ice piles reach especially large volumes in the Arctic seas and impede the operation of the drilling and production platforms and the marine terminals. Therefore, for effective management of the ice conditions within these water areas, the short-term forecast of this phenomenon in accordance with hydrometeorological conditions is necessary.

The mechanism of the ice field's breakdown in contact with a fixed obstacle has been the subject of theoretical studies and models, and is fairly well known at present [1–4]. The drift of the ice cover as a whole and the drift of the single ice fields, their pressure on the obstacle or the collision with it, accompanied by brittle destruction into small ice cakes, which form the pile owing to sinking part of ice cakes to the bottom and rising of rest part to the ice surface, are the obvious and understandable mechanisms of the phenomenon. The intensity of impact with an obstacle or the magnitude of the pressure at the point of contact depends on the drift velocity of the ice cover as a whole and the velocity of a single ice field caused by the wind and the currents.

The compact ice cover drift objectives in the Arctic Basin are also quite well studied, and mathematical models have been developed for this process forecast [5,6]. However, these models are not applicable to local spatial scales of the water area adjacent to the offshore structures, and do not make it possible to predict the characteristics of the ice cover contact with the obstacles and its consequences.

There are investigations of the drift of the single ice field. The main problem is modeling the shape of the ice field, which, in real conditions, is formed under the influence of processes occurring in the drifting ice cover, including collisions, hummocking, freezing and melting. The mathematical description of the movement of the ice floes requires the

parametrization of the impact of the external forces (wind and current) and the resistance to movement. For this purpose, the real irregular shape of the ice floe was simplified and represented by the flat disk for analytical description [7] and by the flat rectangular plate in the experimental studies [8].

The problem of the parametrization of the resistance to movement of objects with irregular shapes can be solved within the theory of ship propulsion [9]. The resistance for the ship movement is divided into separate components, each of which has its own method for calculation. For the friction resistance evaluation, the method of the equivalent flat plate, which has the same area and length as the ship hull, is applied. This approach is suitable for a single ice field with an irregular shape in the plan. When assessing the friction of the wind and the current on the ice field surfaces, the equivalent flat plate approach that has the same surface area and length in the direction of movement can be applied.

This concept was previously used to assess the wind effect on the stationary ice field in contact with the fixed obstacle [10]. Comparison of the generated under the wind action pressure at the point of contact between the ice field and the obstacle with the compressive strength of the sea ice made it possible to estimate the conditions for the ice destruction at the point of contact (the dimensions of the ice field and wind speed), and, as a consequence, the occurrence of the ice pile near the obstacle and its possible volume.

In this article, the same approach is applied to the ice field moving under the action of the wind and the current. The model of the dynamic destruction of the ice field is based on the notion that sea ice, as a solid material, has a certain specific energy of brittle destruction [11–13]. The velocity of the ice field drift determines the kinetic energy of the ice field, which is released when the ice field suddenly stops owing to its collision with an unmovable obstacle. This kinetic energy is mostly realized in the destruction of the ice as material that is in the formation of any fragments—pieces, the volume of which corresponds to the specific energy of the ice destruction and which form the ice pile in the place of collision. It is reasonable to consider that form and dimensions of this ice pile are similar to the ice ridges, which are formed during the collision of drifting ice fields and are quite well studied [14,15].

The paper presents the model of the ice field movement under the action of the wind and current, which is a differential equation that takes into account the effect of wind and current on the ice field, the resistance to movement from air and water masses, as well as initial conditions and a description of the variation over time in the velocity and direction of the wind and current.

This model was applied to the wind drift of an ice field with an arbitrary shape over a stationary water mass. Solving the equation made it possible to estimate its kinetic energy. Based on the assumption that all the kinetic energy at collision with the obstacle and stopping the ice field is realized in the brittle destruction of the ice field, the estimates were made of the potential volume of the ice cakes pile (piling up) dependence on the wind speed and the ice field dimensions.

The results of the study were previously presented at the 26th IAHR International Symposium on Ice [16]. This publication is supplemented by the results of assessing the shape and dimensions of ice piles that can form at the place of a drifting ice field collision with a fixed obstacle.

Research results can be used to predict the formation of ice ridges and piles near offshore structures.

## 2. Problem Statement

The problem under consideration is quite complicated. Therefore, in order to assess the consequences of the collision of an ice field with an obstacle, this process had to be schematized. Three processes are considered, and three tasks (subproblems) are solved, namely, the drift and the destruction of the ice field and then the formation of a pile from the ice fragments. These tasks are solved by different methods, but they are connected in such a way that the results of the first one being solved are the basis for solving the second

task. In order to solve the third task, which is the purpose of the study, the results of solving the second task are used.

The first task is to describe the movement of the ice field under the influence of the external effects: the wind and the currents that act on the ice field, setting it in motion through the friction forces and the velocity pressure on the end surfaces of the ice field. In order to do this, we need to construct the equations of the ice field motion under the action of the wind and the current in some sufficiently large water area that there is no need to consider the influence of the boundaries and the shallow water. The solution of this equation makes it possible to find the speed and the trajectory of the ice field drift under the action of the time-varying wind and the currents and to estimate its kinetic energy.

The solution to the second problem—the assessment of the volume of the ice field fragments as a result of the breakdown under the collision with the fixed obstacle—does not involve the study of the process of the brittle fracture itself, which requires special analysis of the processes occurring in the crystal structure of the sea ice. For the practical application of the under consideration problem, it suffices to estimate "from above", which consists of comparing the kinetic energy of the ice field. This depends on its dimensions and drift velocity, and the specific energy of the brittle destruction of the ice. The main assumption is that all the kinetic energy is converted and spent on the destruction of the ice field. This makes it possible to estimate the potential volume of the ice fragments depending on the wind speed, the ice field dimensions, and its thickness.

There are sufficient grounds to assume that the pattern of formation of the pile from ice fragments hearing an obstacle after collision with the drifting ice field is similar to the patterns of formation of the ice ridges (hummocks) during the collision of the ice fields. This assumption allows us to consider that the configuration (geometric form) of the ice ridge and the ice pile are similar, and on the basis of generalized data on the structure of the ice ridges, to construct a method to estimate the height of the pile (like a hummock sail) above the ice cover.

### 3. Equation of the Ice Field Drift

The movement of the ice field with the specific dimensions and mass under the effect of the wind and the current is considered. It is assumed that the ice field does not contact other ice fields and drifts far enough from the shores of the water area to have no contact with the coastline, and the change in the depth of the sea does not affect the current speed. The wind is assumed to be uniform over the area of the ice field, and its characteristic is the velocity vector, which does not change in its direction and modulus. The same assumption is made for the water flow under the ice field. The friction of the water mass and the atmospheric air on the surface of the ice field occurs in the turbulent regime, and the coefficient of friction is determined by the regularities that were determined for a flat turbulent boundary layer [9,17].

Under the stated assumptions for the coordinate system, where the $OX$ axis is directed along the parallel and the $OY$ axis along the meridian, the equation of the motion of the ice field in the vector form is (Figure 1).

$$(M + m_w)\left(\frac{d\vec{V}}{dt} + f\,\vec{V}\right) = \gamma_a \zeta_{as}\left|\vec{W} - \vec{V}\right|\left(\vec{W} - \vec{V}\right)S_{aw} + \gamma_w \zeta_{ws}\left|\vec{U} - \vec{V}\right|\left(\vec{U} - \vec{V}\right)S_{uw} \qquad (1)$$

In this equation, $M$ is the mass of the ice field, $m_w$ is the added mass of seawater, $S_{aw}$ is the characteristic surface area of the ice field for its part rising above the water surface, $S_{uw}$ is the characteristic surface area of the ice field under the water surface, $V$ is the speed of the ice field motion, $W$—the wind velocity, $U$—the current velocity, $t$—the time, $f$—the Coriolis parameter, $\gamma_a$—the atmospheric air density, $\gamma_w$—the seawater density, $\zeta_a = f_a(W\text{-}V)$—the air resistance coefficient on the ice field surface, depending on the velocity of its drift and the

wind speed, $\zeta_w = f_w(U\text{-}V)$ is the coefficient of the water mass resistance to the movement of the ice field.

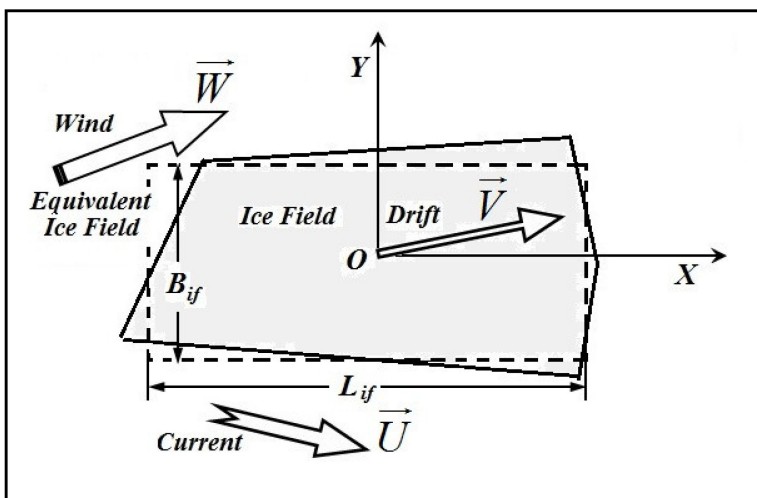

**Figure 1.** Schematic representation of the ice field drifts under the wind and the current action.

The initial conditions for Equation (1) are as follows.

$$t = 0 \ x = 0, \ y = 0, \ \vec{V} = 0$$
$$\vec{W}(0) = \left|\vec{W}(0)\right| \times cos\left(\vec{W}, OX\right), \ \vec{U} = \left|\vec{W}(0)\right| \times cos\left(\vec{U}, OX\right) \tag{2}$$

The equation of the movement should be supplemented with the dependences of the velocity module $f_{W1}$ and the direction $f_{W2}$ of the wind and the velocity module $f_{U1}$ and the direction $f_{U2}$ of the current on time.

$$\left|\vec{W}\right| = f_{W1}(t), \ cos\left(\vec{W}, OX\right) = f_{W2}(t); \ \left|\vec{U}\right| = f_{U1}(t), \ cos\left(\vec{U}, OX\right) = f_{U2}(t) \tag{3}$$

The components included on the right side of Equation (1) characterize the effects of the wind and the current on the ice field. If the wind velocity exceeds the drift velocity of the ice field, that is $(W - V)_{x,y} > 0$, then the wind "moves" the ice field, otherwise: $(W - V)_{x,y} < 0$, the air medium "slows down" the drift of the ice field. The action of the current is similar: at $(U - V)_{x,y} > 0$, the current sets the ice field in motion, and at $(U - V)_{x,y}$, the water mass prevents its drift. The magnitude of the resistance forces depends on the velocity of the ice field relative to the wind velocity and relative to the water surface. The wave resistance is assumed to be negligible due to the relatively low ice drift velocity.

The ice field has an irregular shape, which was formed during the drift as a result of the interaction with other ice fields. It is reasonable to represent the ice field as a rectangular parallelepiped with the following dimensions: $L_{if}$—the length, $B_{if}$—the width and $h_{ice}$—the ice thickness (Figure 1). In order not to consider the orientation of the ice field relative to the trajectory of its motion as the characteristic linear dimension for calculating the friction and the Reynolds number Re, it is reasonable to take the following value, which is close to the average size for the "non-elongated" shape of the ice field in plan and is determined by the following formula ($S_{if}$—the ice field area):

$$L_f = \sqrt{L_{if} B_{if}} = \sqrt{S_{if}} \tag{4}$$

## 4. Steady Wind Drift of the Ice Field

It is considered the drift of the ice field along the stationary water mass under the action of the wind with constant velocity and direction. The curvature of the drift trajectory

due to the Coriolis effect is not taken into account. In this case, the equation of motion (1) takes the following form (the first term is the effect of the wind causing drift, and the second term is the resistance of the water mass to the movement of the ice field).

$$\gamma_a \zeta_{as} (w - v)^2 S_{aw} - \gamma_w \zeta_{ws} v^2 S_{uw} = 0 \tag{5}$$

The coefficients of the air resistance $\zeta_{as}$ and the water one $\zeta_{ws}$ included in the equation consist of two components each: $\zeta^a{}_{vh}$—the head (frontal) resistance of the part of the ice field rising above the water level (the head resistance for the submerged under water part—$\zeta^w{}_{vh}$) and the frictional resistance of the air $\zeta^a{}_{fr}$ and the water $\zeta^w{}_{fr}$ on the flat surfaces of the ice field. The scheme of action of these forces is shown in Figure 2.

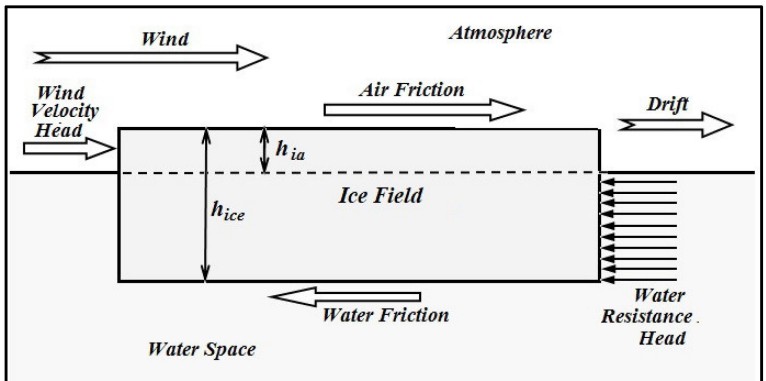

**Figure 2.** Forces that define the wind drift of the ice field.

In this case, it is possible to present the Reynolds number and the coefficient of the friction resistance on the interface atmospheric air—ice with the following form:

$$\text{Re}_{ai} = \frac{w_{sf}}{\nu_a} L_f, \quad \zeta^a_{fr} = \frac{0.455}{(\lg \text{Re}_{ai})^{2.58}} \tag{6}$$

Here, $w_{sf}$—the wind velocity on the upper surface of the ice field, which is determined with respect to the surface rise over the water surface $h_{ia}$, $\nu_a$—the kinematic viscosity of the atmospheric air. Value $h_{ia}$ is connected with the ice field thickness $h_{ice}$ and the ice density $\gamma_{ice,}$ and the seawater density $\gamma_w$ by following the form:

$$h_{ia} = \frac{\gamma_w - \gamma_{ice}}{\gamma_w} h_{ice} \tag{7}$$

Applying known formula for the wind velocity within the boundary layer over the sea surface [18], the following formula was deduced:

$$w_{sf} = w(h_{ia}) = W_{10} \left( 1 + \frac{\sqrt{w}}{\chi_0} \ln \frac{h_{ia}}{z_{10}} \right) \tag{8}$$

In this case, $W_{10}$ ($= w$)—the wind velocity on the height $z_{10} = 10\text{M}$ above the sea level, $\chi_0 = 0.4$—the Karman constant and $c_w$—the coefficient of the sea surface friction. This coefficient depends on the degree of the wave roughness of the sea surface. For the sea surface covered with drifting ice, this coefficient can be estimated by its minimum value $c_w = 1.11 \times 10^{-3}$.

The friction resistance of the ice field on the water mass depends on the drift velocity $v$ and is determined similarly to the formulas for the air resistance (6), that is:

$$\text{Re}_{wi} = \frac{v}{\nu_w} L_f, \quad \zeta^w_{fr} = \frac{0.455}{(\lg \text{Re}_{wi})^{2.58}} \tag{9}$$

Head resistance coefficients can be assumed to be the same for the over-surface and under-surface parts of the ice field. The front and back faces should be considered as the extended barriers, and the following value $\zeta^a{}_{vh} = \zeta^w{}_{vh} = 2.1$ can be accepted [19].

Substituting the obtained results into Equation (4), we get the following transcendental equation, which allows estimating the drift velocity of the single ice field:

$$\gamma_a \left[ \zeta^a_{vh} \, h_{ia} + \zeta^a_{fr} \, L_{if} \right] (w - v)^2 - \gamma_w \left[ \zeta^w_{vh} \, (h_{ice} - h_{ia}) + \zeta^w_{fr} \, L_{if} \right] v^2 = 0 \tag{10}$$

Figure 3 presents the dependence of the ice field (dimension $L_f = 200$ m) drift velocity on the wind velocity for various thicknesses of ice. There is also the dependence of the compact ice cover drift velocity on the wind velocity (the low: 2% of the wind speed) that was established by the observations in the Arctic Ocean [5,6], and the velocity of the wind drift current at the latitude $\varphi = 70°$ N calculated by following formula [19]:

$$v_{wd} = \frac{0.0127 \, W_{10}}{\sqrt{\sin \varphi}} \tag{11}$$

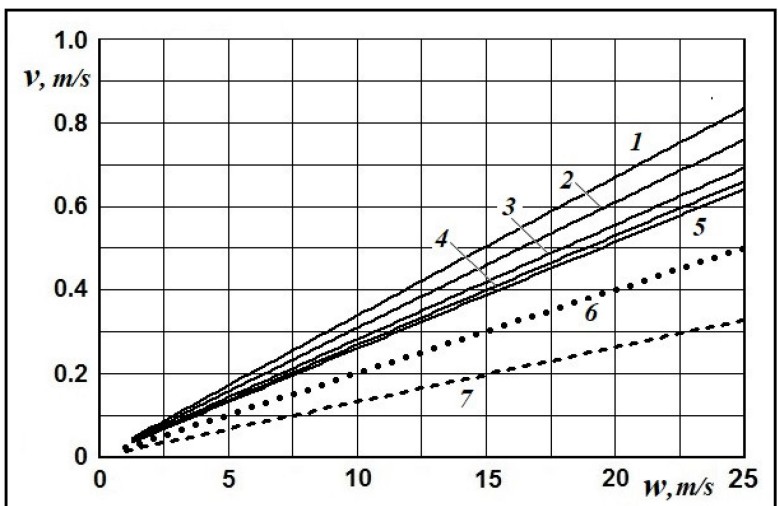

**Figure 3.** Dependence of the ice field drift velocity on the velocity of the wind for various thicknesses of the ice. Notations: *1*—$h_{ice} = 0.25$ m, *2*—$h_{ice} = 0.5$ m, *3*—$h_{ice} = 1$ m, *4*—$h_{ice} = 1.5$ m, *5*—$h_{ice} = 2$ m, *6*—the velocity of the compact ice cover drifting, *7*—the surface wind drift current.

The presented materials show that the velocity of the wind drift of a single ice field significantly exceeds the wind surface current velocity and the velocity of the compact ice cover wind drift in the same water area. These materials also show that the drift velocity of the ice field decreases with increasing ice thickness. It is possible to explain this in the following way: the contribution of the head resistance of the submerged part of the ice field to the total resistance to the wind drift increases as the ice thickness grows.

## 5. Evaluation of the Kinetic Energy of Drifting Ice Field

The value of the kinetic energy of a body is determined by its mass and velocity. For the drifting ice field with thickness $h_{ice}$ and characteristic dimension $L_f$, the kinetic energy is the function of the wind speed $w$ and is given by the following formula:

$$E_{kif}(w) = \frac{1}{2} \gamma_{ice} \, h_{ice} \, L_f^2 \, [v(w)]^2 \tag{12}$$

Expression (12) shows that the kinetic energy of a single ice field increases with the growth of its thickness and area and increases in proportion to the square of the drift velocity *v(w)*. At the same time, the drift velocity of the ice field *v* depends on the dimensions and

the thickness of the ice field. Therefore, to assess the specific nature of this dependence, a special numerical simulation was performed with variations included in (12) parameters by solving Equation (10).

Figure 4 shows the dependence of the kinetic energy of the drifting ice field on the wind speed for various ice thicknesses. The presented results of modeling give the possibility of approximating the dependence of kinetic energy of the ice field on the wind speed by the power dependence:

$$E_{kif}(w) \sim w^{1.97} \tag{13}$$

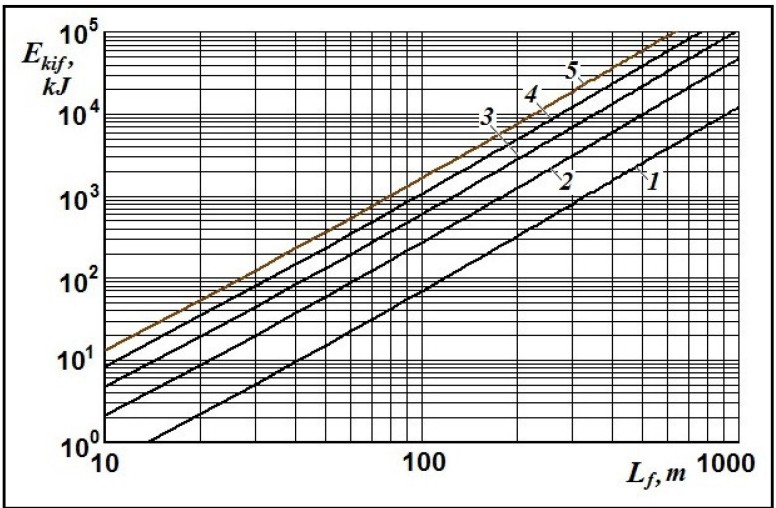

**Figure 4.** Dependence of the drifting ice field kinetic energy on the wind velocity for various thicknesses of the ice. Notations as in Figure 3.

Figure 5 shows the dependence of the kinetic energy of the drifting ice field on its thickness for various wind velocities. These data provide the possibility of approximating the dependence of the kinetic energy on the thickness of the ice field $h_{ice}$ by following the power formula:

$$E_{kif}(h_{ice}) \sim h_{ice}^{0.748} \tag{14}$$

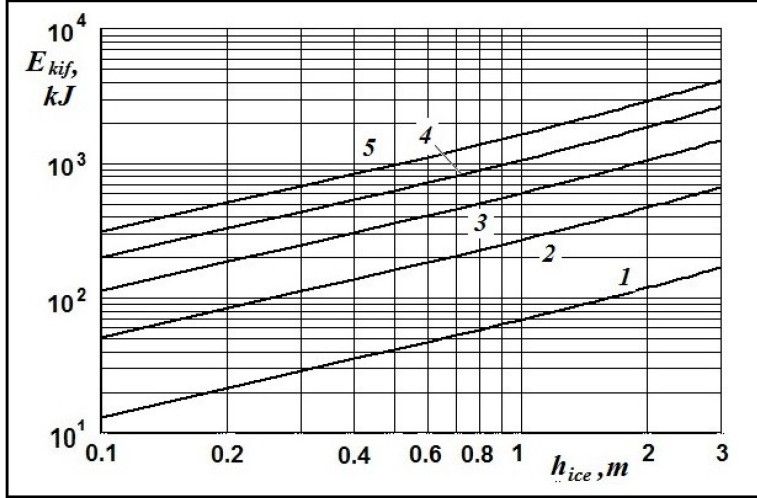

**Figure 5.** Dependence of the drifting ice field kinetic energy on the ice thickness for various wind velocities. Notations: *1*—*w* = 5 m/s, *2*—*w* = 10 m/s, *3*—*w* = 15 m/s, *4*—*w* = 20 m/s, *5*—*w* = 25 m/s.

The results of computations of the dependence of the kinetic energy of the drifting ice field on its characteristic dimension for various wind velocities (presented on a logarithmic scale) show the existence of a power-law dependence between the kinetic energy and dimension. This dependence has the following form:

$$E_{kif}(L_f) \sim L_f^{2.18} \qquad (15)$$

Thus, the multifactorial dependence of the kinetic energy of the drifting ice field on its size and the wind velocity appears, which is contained in Equation (10) implicitly. Formulas (13)–(15), obtained as a result of computer modeling, make it possible to estimate qualitatively the change in the kinetic energy of the drifting ice field with its dimensions and wind speed variation.

## 6. Possible Volume of the Ice Fragments after Collision the Ice Field with an Obstacle

Sea ice is a brittle material; therefore, it fails as a result of mechanical impact, and small ice pieces (fragments, ice cakes) are formed. The ratio of the impact energy and the volume of the formed fragments is characterized by the impact energy of the destruction. According to the available measurement results [11–13], the value of the specific energy of the sea ice destruction lies within the fairly wide range: $e_F = 0.34 \div 4.5 \, \text{kJ/m}^3$. These results were obtained when testing the ice samples that had a salinity from 0 to 6.0‰ and a temperature from $-1\,^{\circ}\text{C}$ to $-60\,^{\circ}\text{C}$.

The obtained solution for the kinetic energy value of the drifting ice field (12) allows us to estimate the volume of the ice destruction (volume of the ice fragments) after the drifting ice field collision with the fixed obstacle. It is assumed the obstacle is absolutely solid, and its length along the waterline exceeds the dimensions of the ice field. The estimation "from above" obtained in this case has the following form:

$$Q_{if}(w) = \frac{\gamma_{ice}}{2\,e_F}\,h_{ice}\,L_f^2\,[v(w)]^2 \qquad (16)$$

By using this formula, it is possible to evaluate the volume of the ice fragments that can appear after the drifting ice field collision for some average value of the specific energy of the ice destruction, for example, $e_F = 1.5 \, \text{kJ/m}^3$. Figure 6 shows the results of calculations: the dependence of the volume of the ice fragments on the wind speed for the ice field with the characteristic dimension $L_f = 200$ m for several ice thicknesses, which are typical for the first-year ice. These materials demonstrate that when the drifting ice field breaks down in the collision with the fixed obstacle, large (hundreds of cubic meters) volumes of the ice fragments can be formed, which will form a pile with significant volume that increases with the ice thickness increasing.

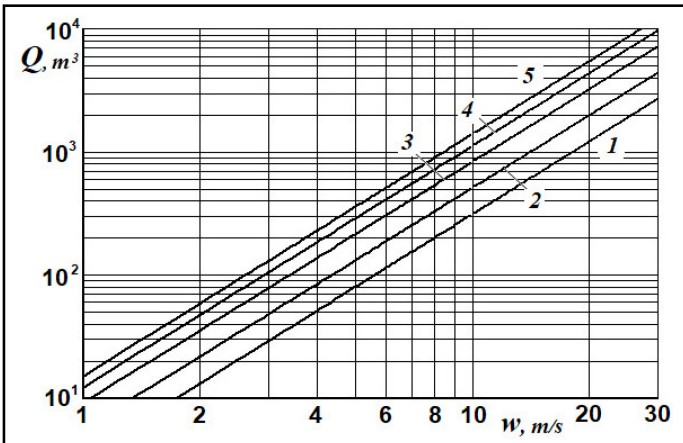

**Figure 6.** Dependence of the ice fragments volume as a result of the ice field crashing upon the wind velocity for various ice thicknesses. Notations as in Figure 3.

The volume of the ice fragments can be comparable to the total volume of the ice field. For example, in Figure 7, we can see for various wind speeds above 20 m/s, the relative volume of the ice fragments may exceed 10% of the pre-collision ice field volume.

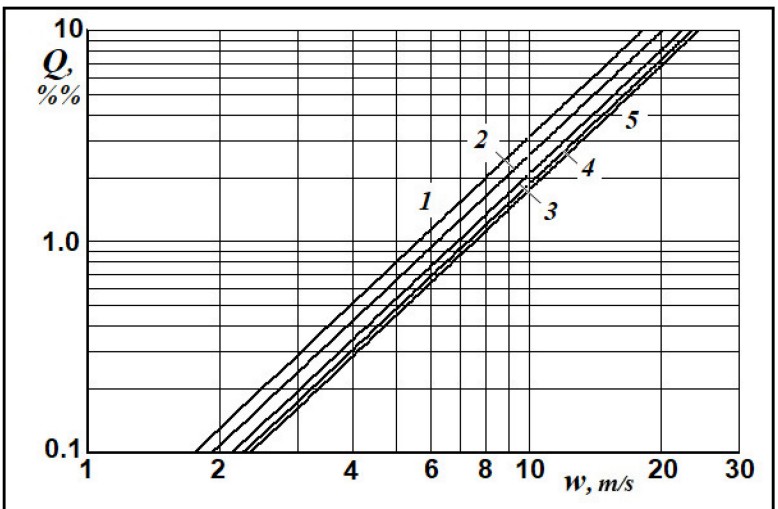

**Figure 7.** Dependence of the ice fragments volume fraction upon the wind velocity for various ice thicknesses. Notations as in Figure 3.

### 7. Estimation of the Possible Dimensions of the Ice Fragments Piles

Accumulations of the ice fragments near the hulls of the stationary drilling and mining platforms make their operation difficult, as the ice piles impede the mooring for the workboats and tankers. The above-stated results of the performed investigations of the wind drift and breakdown of the ice field give possibilities for evaluations of the dimensions of the ice piles near the board of a stationary platform. For this purpose, it is assumed the collision of a drifting ice field with a platform is the main mechanism for the formation of ice accumulation near the boards of the platforms and other offshore structures [4,20,21].

There are sufficient reasons to believe that the ice fragments formed as a result of the collision of a drifting ice field with an obstacle eventually form a pile or ridge, just as it happens when drifting ice fields collide with each other and hummocks and ridges of hummocks are formed.

Therefore, it is possible to apply the results of studying the shape of the hummocks and the ridges of ice fragments to estimate the sizes of the pile of ice fragments in front of the obstacle [14,15,22,23]. The main characteristics that are reasonable to apply to the analysis are the following [15]:

- the ratio of the sail height $H_s$ to the keel depth $H_k$ is equal to $Hk/H_s = \beta_1 = 4.4$,
- the ratio of the keel width $W_k$ to the sail height $H_s$ is equal to $W_k/H_s = \beta_3 = 15.1$,
- the slope angle of the sail $\alpha_s = 20.7°$ and the slope angle of the keel $\alpha_k = 26.6°$.

Figure 8 shows the schema of the pile of ice fragments in front of the obstacle wall, which is half of the ridge, the average cross-section of which is presented in [15]. The adopted schema of the ice pile makes it possible to represent the cross-sectional area of the pile up as the sum of the sail in the form of a triangle and the area of the keel in the form of a trapezoid.

The parameters of the ice pile presented above [15] are sufficient to unambiguously determine the cross-sectional area Aip of the ice section pile in front of the wall, using a single parameter: the sail height $H_s$. Accordingly, the height of the sail of pile $Hs$ can be expressed in terms of the cross-section of pile area $A_{ip}$. The corresponding formula has the following form.

$$H_s = \sqrt{\frac{2\,A_{ip}}{ctg(\alpha_s) + \beta_1[\beta_3 - \beta_1\,ctg(\alpha_k)]}} \tag{17}$$

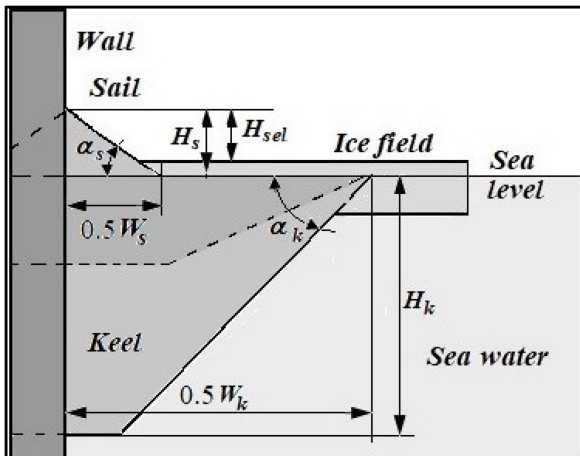

**Figure 8.** Schema of the ice pile in front of an obstacle as the half of the ice ridge (hummock).

In order to estimate the elevation of the pile above the ice cover surface $H_{sel}$, the thickness of the ice field $h_{ice}$ and the density of the sea ice $\gamma_{ice}$ should be taken into account.

$$H_{sel} = \sqrt{\frac{2\,A_{ip}}{ctg(\alpha_s) + \beta_1[\beta_3 - \beta_1\,ctg(\alpha_k)]}} - (1 - \gamma_{ice})\,h_{ice} \qquad (18)$$

The value of the cross-sectional area of the ice pile $A_{ip}$ can be represented as the volume of ice fragments in the following two versions:

1. the volume of the ice fragments per unit length of the pile-up, if the width of the ice field colliding with the obstacle is less than the length of the platform board $L_{pl}$, or
2. the volume of the ice fragments per unit length of the platform board, if the width of the drifting ice field is greater than the length of the platform board.

$$A_{ip} = \frac{Q}{L_f}, \; if \; L_f \le L_{pl} \quad A_{ip} = \frac{Q}{L_{pl}}, \; if \; L_f > L_{pl} \qquad (19)$$

Based on this assumption, the estimations of the height of the ice accumulation near the board of the Prirazlomnaya platform were performed depending on the dimensions of the ice fields and the wind velocity using Formula (18). The computation results are presented in Figure 9.

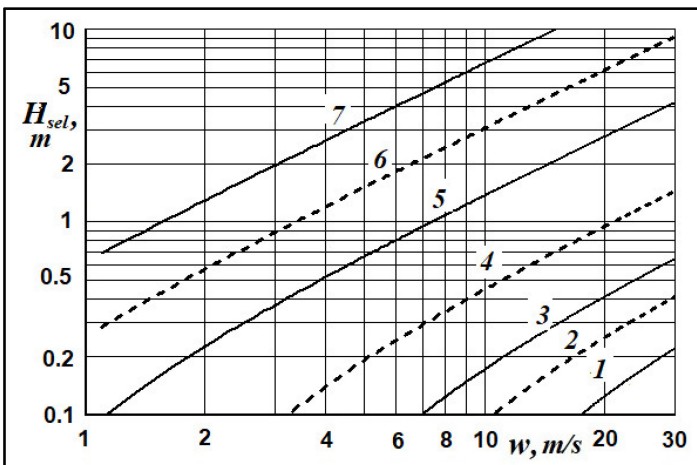

**Figure 9.** Dependence of the ice pile elevation in front of the platform board on the wind velocity and the dimension of the ice field with a thickness of 1 m. Notations: *1*—$L_f$ = 20 m, *2*—$L_f$ = 50 m, *3*—$L_f$ = 100 m, *4*—$L_f$ = 200 m, *5*—$L_f$ = 500 m, *6*—$L_f$ = 1000 m, *7*—$L_f$ = 2000 m.

The estimates of the elevation of the ice pile near the platform board showed the developed model of the ice pile formation made it possible to obtain dimensions close to those observed in real conditions. In particular, by the look of Photo [24], the height of the ice piles observed near the Platform Prirazlomnaya is comparable to the height of the foredeck of the work vessel and to the freeboard of the platform caisson. That is, the height of the ice piles in Photo [24] was about 3 m. Results of modeling in Figure 9 demonstrate that such ice piles could be forming as a result of a single collision with the platform of the ice fields with dimensions of more than $L_f > 500$ m in plan and a thickness of about $h_i = 1$ m, that drifted at a wind speed more than $w = 15$ m/s. These hydrometeorological and ice conditions are observed in the Pechora Sea water area in reality [25].

## 8. Discussion

As a result of the study, the mathematical model of the isolated ice field drift under the influence of the wind and the current was derived, which takes into account the following:

- the ice field mass, its dimensions and the attached water mass,
- the action of the air and the water masses, which, if their velocities exceed the drift speed, set the ice field in motion, and otherwise brake owing to resistance to the ice field motion, and
- the main components of the impact of the air and the water masses on the ice field, parameterized in the form of friction on the upper and lower surfaces of the ice field and in the form of the head pressure (resistance) on the end surfaces of the ice field rising above the water and head resistance (pressure) on the submerged part of its forward surface.

The ice field drift equation was applied to the simplest case of the wind drift of the ice field over the unmovable seawater mass, and modeling of this process was performed, which made it possible to understand the dependence of the ice field drift velocity on the wind speed, its horizontal dimensions and thickness. These results made it possible to estimate the kinetic energy of the ice field and the probable volume of the ice fragments when it collides with the fixed obstacle.

It was assumed that the patterns of the hummocks and ice ridges, on the one hand, and the piles of ice fragments near the stationary platform boards, on the other, as well as their shape and structure, are similar. This made it possible to develop the method and estimate the elevation of the ice piles near the stationary platform board in dependence on the dimension of the ice field and the wind speed.

The height of the ice piles near the Platform Prirazlomnaya was estimated by comparing its dimensions with the height of the foredeck of the work vessel, and the platform's caisson freeboard in the photo turns out to be close to the results of calculations based on the developed ice pile model.

## 9. Conclusions

The performed study is based on the application of methodology developed in the Naval Architect Theory for estimation of the water resistance to the ship hull movement, which has not previously been applied in the studies of the dynamic of the sea ice cover.

The results of estimating the elevations of the piles of ice fragments (cakes) in relation to the Platform Prirazlomnaya are in good agreement with the observed data.

These outcomes provide a basis for future studies, in particular, the ice field drift dynamic under varying wind and tidal currents.

The results of the study (developed models) are recommended to be applied to solve the problems of ice conditions management in the water areas where the platforms and terminals are located in the Arctic seas.

The approach developed approach can be applied to investigations into the consequences of the ice field and the fixed obstacle collisions under the action of other variants of the wind and the current's impact on the ice field.

**Funding:** This research was partially funded by the Ministry of Science and Higher Education of the Russian Federation as part of the World-class Research Center program: Advanced Digital Technologies (contract No. 075-15-2022-312 dated 20 April 2022).

**Institutional Review Board Statement:** Not applicable.

**Informed Consent Statement:** Not applicable.

**Data Availability Statement:** Datas are contained within the article.

**Acknowledgments:** Author expresses his gratitude to the Ministry of Science and Higher Education of the Russian Federation for supporting the studies.

**Conflicts of Interest:** The authors declare no conflict of interest.

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
