# Peer review of "Wind Drift, Breakdown, and Pile Up of the Ice Field"

_jmse, doi:10.3390/jmse11061227_

Round 1

Reviewer 1 Report (Previous Reviewer 1)

The authors have made improvements in this revision. My questions have been answered, and I recommend the article for publication.

Author Response

I would like to express my gratitude for the appreciation of my study.

Reviewer 2 Report (Previous Reviewer 2)

Clearly, I can see that the authors do not modify the manuscirpt much according to the previous suggestion. The quality is still low and not suitable for the journal. So I have to reject it.

The language could be accepted with minor editing.

Author Response

Unfortunately, Reviewer did not provide specific comments either in the first review or in this review. Therefore, there is no way to reasonably respond to comments.

At the same time, I would like to express my gratitude to Reviewer for his attention to my study.

Reviewer 3 Report (Previous Reviewer 3)

I'd like to express my gratitude to the author on responding my previous comments, most of which have been well answered. My only comment for now is the references. The reference for showing the ice conditions in the Pechora Sea (line 398-399) should be provided. And the reference 18 and 24 are not accessible, please correct them.

Author Response

1.     Comment is accepted and reference [25] is added.

2.     References [18] and [24] had been checked and confirmed. At link [18], the book is available for download in *.pdf format.  At link [24], the photograph of the Prirazlomnaya platform with the ice piles is available, but only on the Russian language version of Site.

I would like to express my gratitude for the help in the article preparation.  

Reviewer 4 Report (New Reviewer)

The author spends much time on the manuscript. From his many experiences of the ice behavior research on ice mechanics and ice dynamics, this manuscript is quite valuable for the basic study and the basic scientific reference. Before it can be published, the author should make the following modification.

1 “Figure” in all manuscript should use Figure, not some Fig., and sometimes Figure.

2 Symbols in the formula should give explain. At present, some have and some have not. Of course, the manuscript explains the symbol used XX is YYYYYYYY; XX-YYYYYYY. Please use same style. Do not mixture them. Also some symbols are not same as them in formula.

3 At the end of a formula, it is not necessary to add “.” or “;”.

4 The references are not all follow the MDPI style. They need to modify in same style. In addition to, one of references used Japanese. It is better use web information or doi information instead of the Japanese if you have not the English style of the reference.

The author spends much time on the manuscript. From his many experiences of the ice behavior research on ice mechanics and ice dynamics, this manuscript is quite valuable for the basic study and the basic scientific reference. Before it can be published, the author should make the following modification.

1 “Figure” in all manuscript should use Figure, not some Fig., and sometimes Figure.

2 Symbols in the formula should give explain. At present, some have and some have not. Of course, the manuscript explains the symbol used XX is YYYYYYYY; XX-YYYYYYY. Please use same style. Do not mixture them. Also some symbols are not same as them in formula.

3 At the end of a formula, it is not necessary to add “.” or “;”.

4 The references are not all follow the MDPI style. They need to modify in same style. In addition to, one of references used Japanese. It is better use web information or doi information instead of the Japanese if you have not the English style of the reference.

Author Response

I would like to express my gratitude for the appreciation of my study. Comments are accepted and amendments to text of the article was made.

1.     Only “Figure” is used in all text.

2.     Added notation of symbol for formulas (2), (3) and (4).

3.     “.” and “;” were removed from formula ends.

4.     Reference [24] was aligned with MDPI.

5.     Reference [8]: Japanese title of Journal was translated in English.

This manuscript is a resubmission of an earlier submission. The following is a list of the peer review reports and author responses from that submission.

Round 1

Reviewer 1 Report

The authors present a method of analysis to predict ice pileup and accumulation near floating platforms. Overall this paper and methods will be of interest to the ice modeling community. One item in need of improvement is the English. There are multiple places where the tense of a word need to be improved or 'the' is used incorrectly. (For example, in abstract it should read "Article contains", not "Article contents", "Results of performed the steady" is also not clear. This made the article hard for me to read. I suggest the authors have the article proof-read by someone familiar with English writing.

My summary comments are:

Usually the height of ice above the surface is termed 'freeboard', while the ice below the surface is termed 'draft'. The author uses 'hia' for height above ice. Using the terms freeboard and draft may help simplify notation. in Equation (1) 

Page 4: At the bottom I suggest using word 'schematic' instead of 'schema'. 

Page 5: Text refers to Figure 1 for Lif, Bif, hice, but they are in Figure 2. 

Page 7: Equation (15) I am not clear where the exponent 2.18 came from. The author notes it is a result of "computations of the dependence of kinetic energy of the drifting ice field". Could the author please add more detail on these computations? 

Page 9: The term 'giant huge' is odd. I suggest using either 'giant' or 'huge', but not both together. 

Page 13: reference 18: the website link does not work or does not exist anymore. This should be updated or removed. 

Reviewer 2 Report

This manuscript is an extension of previous study published in IAHR conference. The topic is on the modelling of ice drift under wind and current as well as ice piling against fixed structure. However, it is quite confused to understand how these models are derived and what is novel to readers. The authors should give more explanation on the connection between each model as well. Moreover, the language is a big problem. Many sentences are wrong. To sum up, the manuscript is of low quality. It is not suitable for this Journal. I have to reject it. 

Author Response

No

Comment

Response

1

It is quite confused to understand how these models are derived ….

Explanation.

1.               The model of the ice field wind drift was derived using methodologies developed in the Ship Theory (Naval Architecture methods) and the Aerodynamics. The frictional resistance of the water space (the impact of a wind by friction also) is estimated from the length and area of the submerged part of the hull of vessel, and the frontal resistance of the water (the wind pressure on the front face) was estimated from the velocity in the center of gravity of the frontal surface.

Lines 51 – 58, 221 -223.

2.               The ice fragment formation model is based on the assumption the entire kinetic energy of the drifting ice field, when it stops, is spent on destruction of the ice as a brittle material, and the volume of the ice fragments is determined by the kinetic energy and the specific energy of an ice destruction. This is the “priority from above”. The process of the ice field destruction is not considered, since its specific features cannot exceed this estimate.

Lines 300 – 304.

3.               Modelling of the geometric shape of the pile of the ice fragments in front of the obstacle is based on its external similarity to a hummock formed when the drifting pack ice field collides with a fast ice cover.      

Lines 336 – 338.

2

It is quite confused to understand …. what is novel to readers

Explanation.

1.               A new result for readers is the effective application of the Ship Theory and the Aerodynamics methods for the mathematical description and modelling of the processes occurring in the sea ice cover.

Lines 416 – 418.

2.               The materials of the study of the problem under consideration, presented in the paper (reference [16]), are supplemented by the later derived model for the heap from ice fragments formation and the estimates of its dimensions depending on the ice field sizes and the wind velocity.

Lines 328 – 391.

3

Authors should give more explanation on the connection between each model as well.

Explanation.

Derived models presented in the article are interconnected in the following way. Based on the initial data on the dimensions of the ice field and the wind velocity, the first model estimates the kinetic energy of the drifting ice field. The second model estimates the volume of ice fragments that are formed when the kinetic energy of the ice field released upon collision with an unmovable obstacle. The third model based on this volume determines the shape and sizes of the pile of ice fragments in front of an unmovable obstacle ((that is platform, for example) in dependence on the ice field dimensions and wind speed.

This explanation is included in the text (2. Problem Statement).

Lines 91 – 97.

Reviewer 3 Report

This paper presents a quite simple ice drifting and ice pile up model. The idea is presented properly, but the paper is not well formatted. Significant efforts shall be put on improving the language and the formatting of the paper (e.g., the consistent font size, the resolutions of illustrations, all parameters shall be well explained). Despite of this, the author should also explain why the obstacle size is not playing a role in the Equation 16. And the author claimed that the estimated elevation of the ice pile near the platform board showed that the developed model of the ice pile is close to those observed in real conditions. In my view, this is not a scientific way to validating a numerical model. The author shall show more explicit validation. And the reference style shall be updated to be in line with the journal’s requirements.